

# Large-scale structure of brown rat (*Rattus norvegicus*) populations in England: effects on rodenticide resistance

Mohd Z.H. Haniza[1], Sally Adams[2], Eleanor P. Jones[3], Alan MacNicoll[4], Eamonn B. Mallon[5], Robert H. Smith[6] and Mark S. Lambert[4]

[1] Faculty of Science and Mathematics, Universiti Pendidikan Sultan Idris, Tanjung Malim Perak, Malaysia
[2] School of Life Sciences, University of Warwick, Coventry, United Kingdom
[3] Fera Science Ltd., York, United Kingdom
[4] Animal and Plant Health Agency, York, United Kingdom
[5] Department of Genetics, University of Leicester, Leicester, United Kingdom
[6] School of Applied Sciences, University of Huddersfield, Huddersfield, United Kingdom

Corresponding author
Mark S. Lambert,
mark.lambert@apha.gsi.gov.uk

## ABSTRACT

The brown rat (*Rattus norvegicus*) is a relatively recent (<300 years) addition to the British fauna, but by association with negative impacts on public health, animal health and agriculture, it is regarded as one of the most important vertebrate pest species. Anticoagulant rodenticides were introduced for brown rat control in the 1950s and are widely used for rat control in the UK, but long-standing resistance has been linked to control failures in some regions. One thus far ignored aspect of resistance biology is the population structure of the brown rat. This paper investigates the role population structure has on the development of anticoagulant resistance. Using mitochondrial and microsatellite DNA, we examined 186 individuals (from 15 counties in England and one location in Wales near the Wales–England border) to investigate the population structure of rural brown rat populations. We also examined individual rats for variations of the *VKORC1* gene previously associated with resistance to anticoagulant rodenticides. We show that the populations were structured to some degree, but that this was only apparent in the microsatellite data and not the mtDNA data. We discuss various reasons why this is the case. We show that the population as a whole appears not to be at equilibrium. The relative lack of diversity in the mtDNA sequences examined can be explained by founder effects and a subsequent spatial expansion of a species introduced to the UK relatively recently. We found there was a geographical distribution of resistance mutations, and relatively low rate of gene flow between populations, which has implications for the development and management of anticoagulant resistance.

## INTRODUCTION

The brown rat (*Rattus norvegicus*) first arrived in Britain in the early 18th century, originally from the steppes of Central Asia (*Yalden, 1990*), and is regarded as one of the most important vertebrate pest species in the UK (*Buckle & Smith, 2015*). Brown rats

are associated with risks to public health, animal health, and impacts on agriculture, infrastructure and native wildlife (*Meehan, 1984*; *Webster, Ellis & MacDonald, 1995*). Anticoagulant rodenticides have been widely used for controlling rat populations in the UK since the introduction of warfarin in the 1950s (*Hayes, 1950*). However, resistance to warfarin was encountered by 1960 (*Boyle, 1960*), and it is thought that chance genetic mutations arise that confer heritable resistance to anticoagulant compounds; intensive use of anticoagulants applies a selection pressure that increases the frequency of resistant rats within a population (*Greaves, 1995*; *Greaves & Ayres, 1967*; *Smith & Greaves, 1987*). In response to the emergence of warfarin resistance, 'second-generation' anticoagulant rodenticides, including bromadiolone and difenacoum, were developed that controlled warfarin-resistant rats (*Hadler & Shadbolt, 1975*; *Hadler, Redfern & Rowe, 1975*), although some rat populations have now evolved resistance to some of these more toxic anticoagulants (*Lund, 1985*; *Redfern & Gill, 1978*).

Warfarin acts by interfering with blood coagulation (blood clotting). Several factors in the coagulation process are dependent on sufficient vitamin K levels for their proper functioning (*Suttie, 1980*). Vitamin K hydroquinone is an essential cofactor for post-translational gamma-carboxylation of these blood coagulation factors (*Sadowski, Esmon & Suttie, 1976*). During each carboxylation step, one molecule of vitamin K hydroquinone is oxidized to vitamin K 2,3 epoxide. The recycling of this micronutrient is carried out by the vitamin K epoxide reductase (VKOR) complex. Suppression of the VKOR by anticoagulants inhibits the carboxylation of clotting factors and thus compromises the coagulation process (*Bell & Caldwell, 1973*). Early studies showed that warfarin resistance was inherited in rats as a single autosomal dominant gene (Rw) (*Greaves & Ayres, 1967*). More recently it has been demonstrated that mutations or variations (Single Nucleotide Polymorphisms) in a gene of the VKOR complex, *VKORC1*, are involved in the resistance to anticoagulants in rats (*Pelz et al., 2005*; *Rost et al., 2004*). These variations, which result in amino acid substitutions in the protein VKORC1, may decrease the sensitivity of the protein to warfarin, maintaining the efficiency of the coagulation process in warfarin-exposed resistant rats (*Pelz et al., 2005*).

It has been suggested that rodenticide resistance spread in European rat populations from initial focal points, with resistant rats expanding into new areas (*Pelz et al., 2005*). Earlier work noted that resistance spread at about 5 km per year (*Brodie, 1976*; *Drummond, 1970*). However, warfarin resistance, though widespread, is not ubiquitous. One possible explanation for this pattern is that gene flow between different rat populations is limited. The rate at which resistance is acquired is a function of the resistance allele's frequency, its dominance, the relative fitness of being resistant and, often overlooked, the pest organism's population structure (*Roush & McKenzie, 1987*). Population structure, the subdivision of populations into smaller interbreeding units, is important as it controls gene flow from area to area. Gene flow has two effects on the spread of resistance. Firstly, the greater the gene flow between areas, the more likely resistance genes are to spread. Secondly, if resistance genes spread into areas where pesticides are not used, resistance genes may be diluted by susceptible individuals (*Wool & Noiman, 1983*). In this study, we

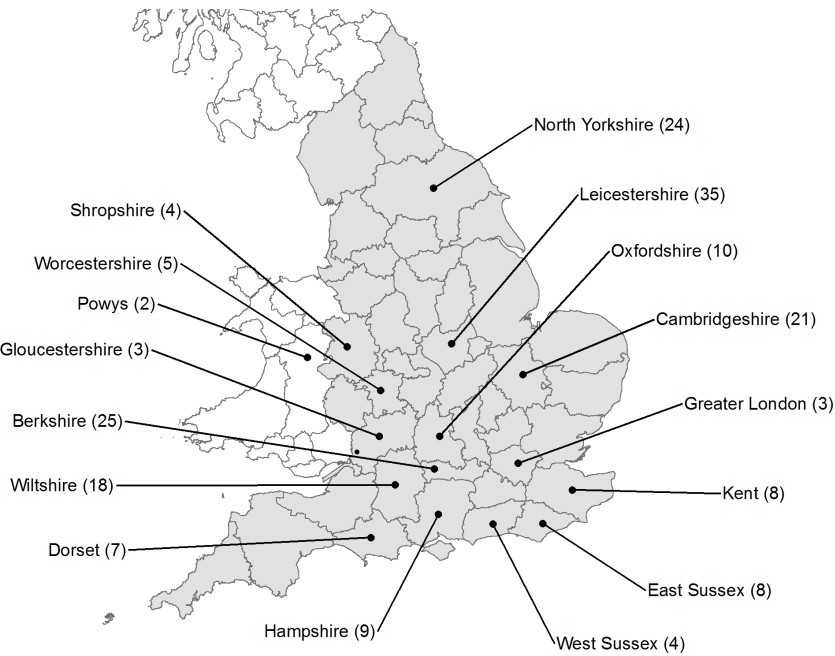

**Figure 1 Geographical distribution of samples.** Current (ceremonial) counties shown for England (shaded), Wales and Scotland; place markers denote county centroids rather than exact origin of samples. Contains Ordnance Survey data (©) Crown Copyright 2015.

used microsatellites, which are informative of population level structures and gene-flow between populations (e.g., *Balloux & Lugon-Moulin, 2002*; *Hutchison & Templeton, 1999*), and mitochondrial DNA sequences, which can provide information about older events in the colonisation history of the rats in the UK, as has been done for example, for house mice (*Searle et al., 2009*).

We examined 186 rural brown rat DNA samples (184 from England and two from a location in Wales near to the Wales–England border) in order to investigate population structure through mitochondrial DNA (mtDNA) and microsatellite analysis. We also examined the distribution of VKORC1 variants within the sampled populations. The samples were collected opportunistically and hence sampling was not geographically stratified, nor was regional coverage comprehensive; however, the results reveal novel insights into the structure of rural brown rat populations that have important implications for the development and management of anticoagulant resistance.

## MATERIALS AND METHODS

### Sample details

We analysed 184 rats from 15 counties in England; we also included two rats collected from Welshpool, Powys, 6 km from the Wales–England border (Fig. 1). The majority of the samples came from an extensive trapping program carried out by the Animal and Plant Health Agency's (APHA) National Wildlife Management Centre (then part of the Central Science Laboratory), York from 1990–2000. These samples were obtained as livers stored at −20 °C. Rats were also trapped on several farms in Leicestershire and Yorkshire during

2004 and 2005 and stored at −20 °C. In addition, samples from Cambridge were obtained from rats trapped in 2004 as extracted DNA from the Babraham Institute, Cambridge. All research that involved the use of live animals was approved in advance by written confirmation from the APHA's Animal Welfare and Ethical Review Body (AWERB) or equivalent at the time of the study. Our AWERB membership has both internal and external members including lay representatives and experts in a variety of apposite areas such as veterinary surgery, statistics and animal welfare.

## DNA extraction

Total genomic DNA was extracted from either 20 µg of liver or a 15 mm tail tip using the Wizard SV Genomic DNA Purification System (Promega, Madison, WI, USA) as per instructions. DNA samples were stored at −20 °C until use.

## Mitochondrial DNA (mtDNA)

For each individual, a 425 base pair region of the hypervariable region 1 (HVR1) segment within the mtDNA control region was PCR amplified. PCR was carried out as per *Hingston et al. (2005)* with slight modifications. Primers used were L283 (5′-TACACTGGTCTTGTAAACC-3′) and H16498 (5′-CCTGAAGTAGGAACCAGATG-3′). A 20 µl reaction was used in which 2 µl genomic DNA was added to the reaction mixtures containing 10 µl PCR reaction mix (YorkBio, York, UK), 1 µl (10 uM) of each primer and 6 µl of $H_2O$. PCR was carried out on a T1 Thermocycler (Biometra, Goettingen, Germany). The PCR conditions were: 30 cycles of denaturation at 94 °C for 45 s, annealing at 50 °C for 45 s, elongation at 72 °C for one min, and a final extension at 72 °C for 30 min.

PCR products were then cleaned with the YorkBio PCR Cleanup kit to remove any unincorporated nucleotides and primers that can interfere with the sequencing process. The cleaned PCR products were sequenced by the John Innes Genome laboratory using an ABI3700 capillary sequencer.

## Microsatellite DNA

We used six microsatellite loci from the literature, that are highly variable for brown rats (D3, D5, D8, D12, D16, D17) (*Heiberg, 2002*). The PCR volume was 10 ul, containing 5 ul of PCR reaction mix (YorkBio), 0.5 ul of DNA, 0.5 ul of each primer and 3.5 ul of $ddH_2O$. One primer of each locus was labelled with one of the fluorescent dyes, PET, 6FAM or VIC. PCR was run on a T1 Thermocycler (Biometra, Goettingen, Germany) using the following conditions: one cycle of denaturation at 94 °C for five mins, 35 cycles of denaturation (94 °C for 30 s), annealing (50 °C for 30 s) and elongation (72 °C for 30 s). Finally, there was one cycle of extension at 72 °C for 30 s. PCR products were run as two batches on an ABI 3730 capillary sequencer (Applied Biosystems, Foster City, CA, USA). The results were scored using GeneMapper v0.5.0 (Applied Biosystems, Foster City, CA, USA).

## VKOR analysis

*Pelz et al. (2005)* identified seven different VKORC1 variants in warfarin resistant Norway rats; four of these (Tyr139Cys, Leu120Gln, Leu128Gln and Tyr139Ser) were reported from

**Table 1 Variations in exon 3 of VKORC1 (after *Pelz et al., 2005*).** The variations are labeled as wild-type/position/mutant, so Tyr139Cys is a tyrosine at the 139th amino acid converted to a cysteine in the variant.

| Mutation | Detection method | Wildtype | Mutant |
| --- | --- | --- | --- |
| Tyr139Cys | ARMS-PCR | 168 bp | 168 bp |
| | | 123 bp | 101 bp |
| Leu120Gln | *Stu* I | 330 bp | 195 bp |
| | | | 135 bp |
| Leu128Gln | *Bsr* I | 330 bp | 170 bp |
| | | | 160 bp |
| Tyr139Ser | *Mnl* I | 160 bp | 110 bp |

the United Kingdom (Table 1), we therefore analysed our samples for the presence of these four variants. A sub-sample of the rats had previously been screened for resistance to warfarin using a blood clotting response (BCR) test; individuals which were determined by the BCR test to be warfarin resistant ('BCR positives') but were found to have none of the above VKORC1 variants were screened for other VKOR polymorphisms by genetic sequencing. PCR primers and detailed protocols were as per *Pelz et al. (2005)*. All PCR products were visualized on a 3% agarose gel.

### ARMS-PCR

Amplification refractory mutation system (ARMS)-PCR was used to detect the Tyr139Cys variant. This technique employs two primer pairs to amplify, respectively, the two different alleles of a single nucleotide polymorphism (SNP) in a single PCR reaction.

### Restriction fragment length polymorphisms

The other three VKORC1 variants create novel restriction sites in exon 3, which allow one of the three enzymes (*Stu* I, *Bsr* I, *Mnl* I) to cut the fragment at this point creating fragments of characteristic sizes (Table 1).

## Data analysis

For the mtDNA analyses, East Sussex and West Sussex were combined (such that for Sussex, $n = 12$), Shropshire and Powys were combined (such that for Shropshire + Powys $n = 6$). We analysed the phylogenetic relationship among the mtDNA haplotypes using two methods. First, a minimum spanning network based on a matrix of the observed nucleotide differences was calculated using the program ARLEQUIN 3.01 (*Excoffier, Laval & Schneider, 2005*). Second, the genetic distance between haplotypes was calculated assuming a *Tamura & Nei (1993)* model of sequence evolution. These distances were then used to construct a neighbour-joining tree using the computer program SPLITSTREE (*Huson, 1998*); 1,000 bootstrap replicates were calculated to estimate the support for each node in both the minimum spanning network and the neighbour-joining tree. For comparison, we included Genbank sequences from several laboratory rat strains (Wistar, BN/SsNHsdMCW, F344 X BN F1 and Sprague–Dawley), a wild caught brown rat from Denmark, and as an outgroup, a *Rattus rattus* sequence.

The geographic distribution of genetic variation was estimated using Analysis of Molecular Variance (AMOVA) performed by ARLEQUIN 3.01. Gene diversity ($h$) and nucleotide diversity ($\pi$) of the various populations and their respective standard deviations were calculated using ARLEQUIN 3.01. ARLEQUIN 3.01 was also used to perform mismatch analysis to compare the distribution of the observed number of differences between pairs of haplotypes (mtDNA) and the expected distribution under various models of population change (*Rogers & Harpending, 1992*).

Geographical distances were calculated as the distance from the central national grid reference of one population area to the central national grid reference of the other. This distance was compared with $\Phi_{st}$, an analogue of Wright's $F_{st}$ statistic (*Wright, 1951*), using a Mantel test carried out with the computer program GenAlEx 6 (*Peakall & Smouse, 2006*). Observed and expected heterozygosities were calculated for the microsatellite data in ARLEQUIN 3.01.

For the microsatellite analyses, East Sussex and West Sussex were combined (such that for Sussex, $n = 12$), Powys and Shropshire were combined ($n = 6$); samples from Greater London ($n = 3$) and Gloucestershire ($n = 3$) were not included (because of the small sample sizes). Using microsatellite data, inferences on the number of populations were made with the fully Bayesian clustering method implemented in STRUCTURE 2.1 (*Pritchard, Stephens & Donnelly, 2000*). With the aim of determining the most likely number $K$ of population units, the program was run ten times for $K = 2$ to $K = 10$. The model with admixture has been used with correlated frequencies. After some preliminary tests of the convergence time needed for the Monte–Carlo Markov chain, a burn-in period of 100,000 steps followed by 1,000,000 steps was used. The most likely value of $K$ was considered using the maximum log likelihood values for the aggregated runs, and by considering Delta $K$, the rate of change in the log-likelihood values between the values of $K$ (*Evanno, Regnaut & Goudet, 2005*), calculated in Structure Harvester (*Earl & vonHoldt, 2012*). Structure plots were created for $K = 3$–$5$ using the programs CLUMPP (*Jakobsson & Rosenberg, 2007*) and DISTRUCT (*Rosenberg, 2004*), which aggregate and plot out the multiple runs for each value of $K$ into single outputs.

## RESULTS

### Molecular diversity

A total of six unique mtDNA haplotypes were represented in the individuals sampled. The nucleotide differences between haplotypes are shown in Table 2. Sequences representing each unique mtDNA haplotype have been deposited in Genbank under accession numbers DQ897633–DQ897638. There are 10 variable nucleotide positions all of which are transitions (Table 2), forming two haplogroups. These are RNH1, the most common haplotype, and RNH6, which diverges from it by a single mutation, RNH2, the second most common haplotype, and RNH5 and RNH3, which diverge from it by single mutations with a further haplotype RNH3, two mutations from RNH5 (Figs. 2A and 2B). Figure 2B also includes several other sequences obtained from Genbank for this region of the *R. norvegicus* mitochondrial genome. These include sequences from several

**Table 2 The nucleotide differences of the 6 mtDNA haplotypes (RNH1-RNH6) in a sample of 185** *Rattus norvegicus* **individuals from 15 different sampling areas in the United Kingdom.** The top row is the position of the variable nucleotides within the 425 bp sequence.

| Position | 95 | 97 | 157 | 204 | 244 | 246 | 260 | 265 | 276 | 313 | *n* |
|---|---|---|---|---|---|---|---|---|---|---|---|
| Haplotype | | | | | | | | | | | |
| RNH1 | T | T | C | T | C | T | T | G | A | G | 130 |
| RNH2 | . | C | . | C | . | C | C | A | G | A | 49 |
| RNH3 | . | C | T | C | T | C | . | A | G | A | 3 |
| RNH4 | C | C | . | C | . | C | C | A | G | A | 1 |
| RNH5 | . | C | . | C | . | C | . | A | G | A | 1 |
| RNH6 | . | . | . | . | . | . | . | . | . | A | 1 |

laboratory rat strains (Wistar, BN/SsNHsdMCW, F344 X BN F1 and Sprague-Dawley), a wild caught brown rat from Denmark, and as an outgroup, a *Rattus rattus* sequence. Relative to our samples, the Wistar strain has a deletion at position 77, the *R. rattus* and Sprague-Dawley strain have an insertion at position 305. The *R. rattus* sample also has an insertion at position 266. Neither haplogroup (nor haplotype) dominated any region, nor was there any geographic structuring to the haplotype distributions. The haplotypic (*h*) and nucleotide diversity (*π*) indices are given in Table 3. These values range from 0 (monomorphic populations in Leicestershire, Dorset, Hampshire and Greater London) to $h = 0.67$ and $\pi = 0.0118$ in Gloucestershire, with an overall average of $0.44 \pm 0.03$ for *h* and $0.0069 \pm 0.0040$ for *π*. Figure 3 shows the observed number of differences between pairs of haplotypes (mismatch analysis). This distribution is not significantly different from the spatial expansion model; $SSD = 0.1023$, Bootstrap replicates $= 1{,}000$, $p = 0.133$ (*Rogers & Harpending, 1992*).

Eighty-one alleles were found across six microsatellite loci tested in all individuals. All of the loci were polymorphic and the number of alleles ranges from 10 (D5) to 18 (D12). The mean number of alleles detected at each locus was 13.5. The observed heterozygosities (Ho) and expected heterozygosities (He) are shown in Table 4. The highest observed heterozygosity was seen in locus D8 in the Dorset population and locus D3 in the Worcester population, and the lowest was locus D5 in the Worcester population. Only 22.6% of the Ho were higher than He, 1.2% of Ho were the same as He (locus D5 from the Wiltshire population) and 76.2% of Ho were lower than He. A substantial deficit of heterozygotes was observed for locus D17 in the Berkshire, Leicestershire, Wiltshire and Yorkshire populations, locus D5 in the Sussex and Worcestershire populations, and locus D8 in the Worcestershire population.

## Genetic structure
### Mitochondrial DNA

A result was not obtained for one sample from Kent; hence 185 samples were included in the analyses. AMOVA ($p < 0.001$) showed that 33.4% of the total variance was assigned to between population (county) diversity ($df = 13$, sum of squares $= 105.9$, variance

A)

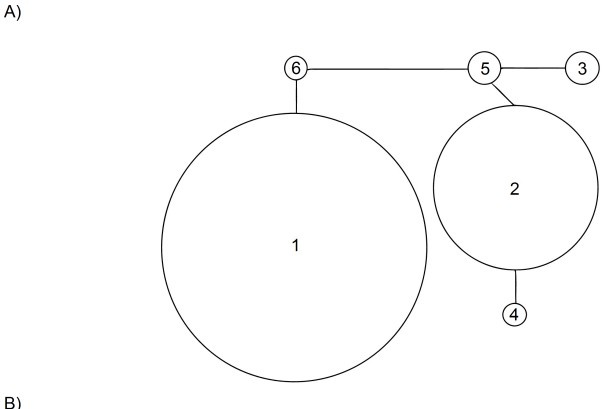

B)

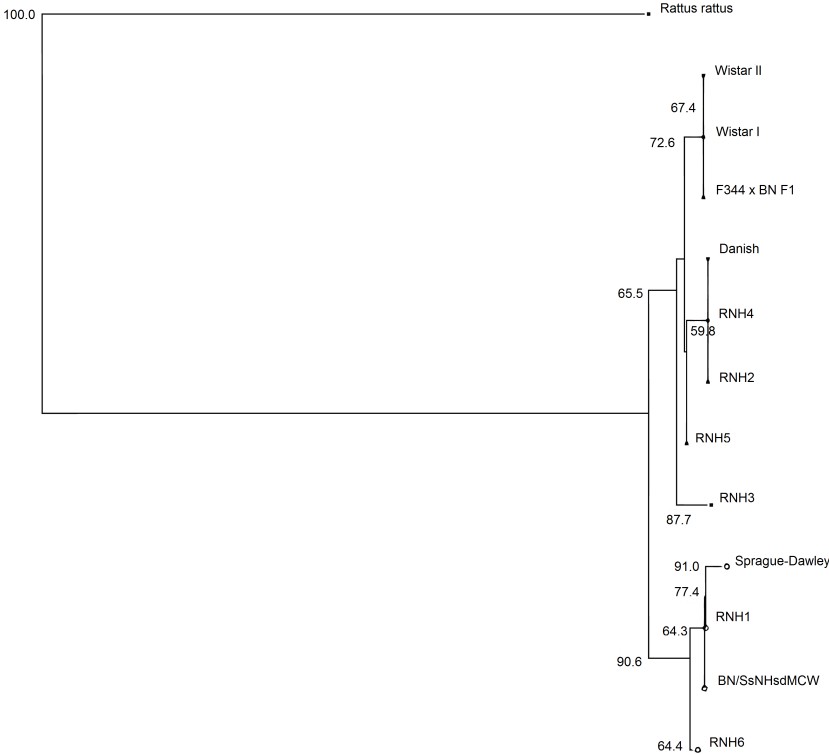

**Figure 2 Phylogenetic relationships from mtDNA; study samples and closely-related Genbank sequences.** (A) Minimum spanning network of the phylogenetic relationships between the 6 mtDNA haplotypes found. The area of the circles represents the frequency of the haplotypes in the entire population. (B) Neighbour-joining tree calculated with *Tamura & Nei (1993)* distances for the 6 mtDNA haplotypes of English brown rats (RNH1-6), several strains of lab rats (Wistar—Accession numbers: MIRNXX, RNMITDLO, Sprague-Dawley—MIRNDNC, BN/SsNHsdMCW–AY172581, F344 X BN F1–AY769440), a Danish wild caught brown rat—RNO428514 and the closely related black rat *R. rattus*–DQ009794. The percentage bootstrap support (1,000 replicates) are shown for nodes with greater than 50% support.
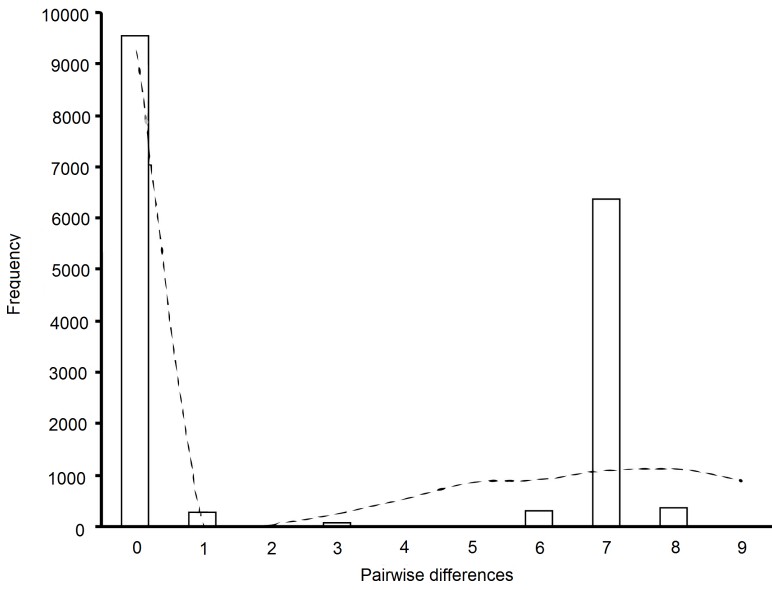

**Figure 3 Mismatch distribution of pairwise sequence differences in brown rat mtDNA (bars).** The dotted line represents the expected results from a spatial expansion model.

**Table 3 Haplotype ($h$) and nucleotide diversity ($\pi$) and their standard deviations (SD) based on mtDNA of the populations sampled.**

| Area | $n$ | $h \pm$ SD | $\pi \pm$ SD |
|------|-----|-----------|--------------|
| Total | 185 | $0.44 \pm 0.03$ | $0.0069 \pm 0.0040$ |
| Berkshire | 25 | $0.49 \pm 0.09$ | $0.0077 \pm 0.0046$ |
| Cambridgeshire | 21 | $0.27 \pm 0.12$ | $0.0043 \pm 0.0029$ |
| Dorset | 7 | $0.00 \pm 0.00$ | $0.0000 \pm 0.0000$ |
| Gloucestershire | 3 | $0.67 \pm 0.31$ | $0.0118 \pm 0.0098$ |
| Greater London | 3 | $0.00 \pm 0.00$ | $0.0000 \pm 0.0000$ |
| Hampshire | 9 | $0.00 \pm 0.00$ | $0.0000 \pm 0.0000$ |
| Kent | 7 | $0.57 \pm 0.12$ | $0.0101 \pm 0.0065$ |
| Leicestershire | 35 | $0.00 \pm 0.00$ | $0.0000 \pm 0.0000$ |
| North Yorkshire | 24 | $0.51 \pm 0.05$ | $0.0089 \pm 0.0052$ |
| Oxfordshire | 10 | $0.53 \pm 0.10$ | $0.0094 \pm 0.0058$ |
| Shropshire + Powys | 6 | $0.60 \pm 0.21$ | $0.0123 \pm 0.0085$ |
| Sussex | 12 | $0.41 \pm 0.13$ | $0.0072 \pm 0.0045$ |
| Wiltshire | 18 | $0.31 \pm 0.13$ | $0.0049 \pm 0.0032$ |
| Worcestershire | 5 | $0.40 \pm 0.24$ | $0.0071 \pm 0.0052$ |

components $= 0.55$) while 66.6% was attributed to diversity within populations ($df = 170$, sum of squares $= 186.02$, variance components $= 1.09$).

### Microsatellite DNA

A result was not obtained for one sample from Sussex; hence 179 samples were included in the analyses (Greater London and Gloucestershire were not included). AMOVA

**Table 4  Microsatellite alleles.**

| County | Number of alleles in each locus | | | | | | Mean number of alleles | Heterozygosity (observed/expected) | | | | | |
|---|---|---|---|---|---|---|---|---|---|---|---|---|---|
|  | D17 | D3 | D5 | D12 | D14 | D8 |  | D17 | D3 | D5 | D12 | D14 | D8 |
| Berkshire | 8 | 9 | 5 | 15 | 9 | 11 | 9.67 | 0.32/0.74 | 0.76/0.83 | 0.44/0.71 | 0.84/0.90 | 0.68/0.85 | 0.60/0.90 |
| Cambridge | 5 | 8 | 6 | 12 | 6 | 11 | 8.00 | 0.38/0.36 | 0.85/0.80 | 0.76/0.72 | 0.76/0.85 | 0.62/0.70 | 0.71/0.83 |
| Dorset | 6 | 7 | 5 | 5 | 5 | 5 | 5.50 | 0.71/0.83 | 0.71/0.82 | 0.85/0.80 | 0.43/0.77 | 0.57/0.86 | 1.00/0.78 |
| Hampshire | 6 | 5 | 2 | 7 | 6 | 5 | 5.16 | 0.77/0.84 | 0.77/0.78 | 0.44/0.50 | 0.66/0.83 | 0.66/0.80 | 0.55/0.67 |
| Kent | 8 | 6 | 2 | 9 | 7 | 7 | 6.50 | 0.75/0.85 | 0.87/0.83 | 0.50/0.40 | 0.62/0.91 | 0.87/0.89 | 0.62/0.90 |
| Leicestershire | 9 | 8 | 6 | 10 | 11 | 12 | 9.33 | 0.31/0.70 | 0.74/0.80 | 0.60/0.64 | 0.71/0.76 | 0.83/0.87 | 0.77/0.88 |
| North Yorkshire | 9 | 10 | 5 | 10 | 10 | 12 | 9.33 | 0.31/0.68 | 0.79/0.82 | 0.37/0.64 | 0.79/0.86 | 0.62/0.86 | 0.79/0.90 |
| Oxfordshire | 7 | 6 | 2 | 9 | 8 | 8 | 6.67 | 0.50/0.83 | 0.70/0.79 | 0.40/0.48 | 0.50/0.92 | 0.70/0.88 | 0.70/0.90 |
| Shropshire + Powys | 4 | 6 | 5 | 7 | 6 | 6 | 5.67 | 0.33/0.45 | 0.83/0.86 | 0.50/0.79 | 0.50/0.83 | 0.66/0.87 | 0.50/0.86 |
| Sussex | 7 | 6 | 4 | 10 | 7 | 8 | 7.00 | 0.45/0.69 | 0.72/0.85 | 0.27/0.61 | 0.72/0.90 | 0.72/0.75 | 0.63/0.87 |
| Wiltshire | 6 | 9 | 5 | 11 | 7 | 10 | 8.00 | 0.22/0.61 | 0.72/0.78 | 0.66/0.66 | 0.83/0.78 | 0.66/0.77 | 0.61/0.81 |
| Worcester | 3 | 5 | 2 | 5 | 4 | 6 | 4.16 | 0.60/0.68 | 1.00/0.82 | 0.00/0.53 | 0.80/0.87 | 0.40/0.77 | 0.20/0.91 |

($p < 0.010$) showed that 8% of the total variance was assigned to between population diversity ($df = 11$, sum of squares = 141.25, variance components = 0.50) while 92% was assigned to diversity within populations ($df = 167$, sum of squares = 952.44, variance components = 5.70). As the county structure did not explain much of the variation found in the microsatellite data, we clustered the samples with STRUCTURE. The optimal number for $K$ was assessed both by the maximum log-likelihood, which reached a plateau between $K = 3$ and $K = 5$, and by Delta $K$, which showed a clear spike at $K = 3$ (Fig. 4A for Delta $K$). These three populations were not geographically clear cut, although there was a 'northern' (Yorkshire, Cambridgeshire), a 'western' (Dorset, some of the Shropshire and Welsh border populations, Wiltshire) and 'central/eastern' (Leicester, Oxford, Sussex, Worcester) cluster. The remaining populations (Berkshire, Hampshire, Kent) were admixed in approximately equal proportions of the 'eastern' and 'central/eastern' clusters. At $K = 4$, the 'central/eastern' cluster forms two clusters and some of the admixed populations become less admixed. At $K = 5$, the 'central/eastern' cluster breaks down even further but the populations are less resolved (Fig. 4B).

### Geographical distance

A Mantel test found no association between $\Phi_{st}$ (based on mtDNA results) and the geographical distance between the populations ($r = 0.171$, $p = 0.120$, $n = 91$). To examine whether variance increased over geographical distance, the residuals from a linear regression of $\Phi_{st}$ against geographical distance were plotted against geographical distance (*Hutchison & Templeton, 1999*); a Mantel test found no significant relationship ($r = 0.0001$, $p = 0.482$, $n = 91$). Similarly, the microsatellite DNA results showed no difference between $F_{ST}$ and the geographical distance ($r = 0.215$, $n = 91$, $p = 0.150$).

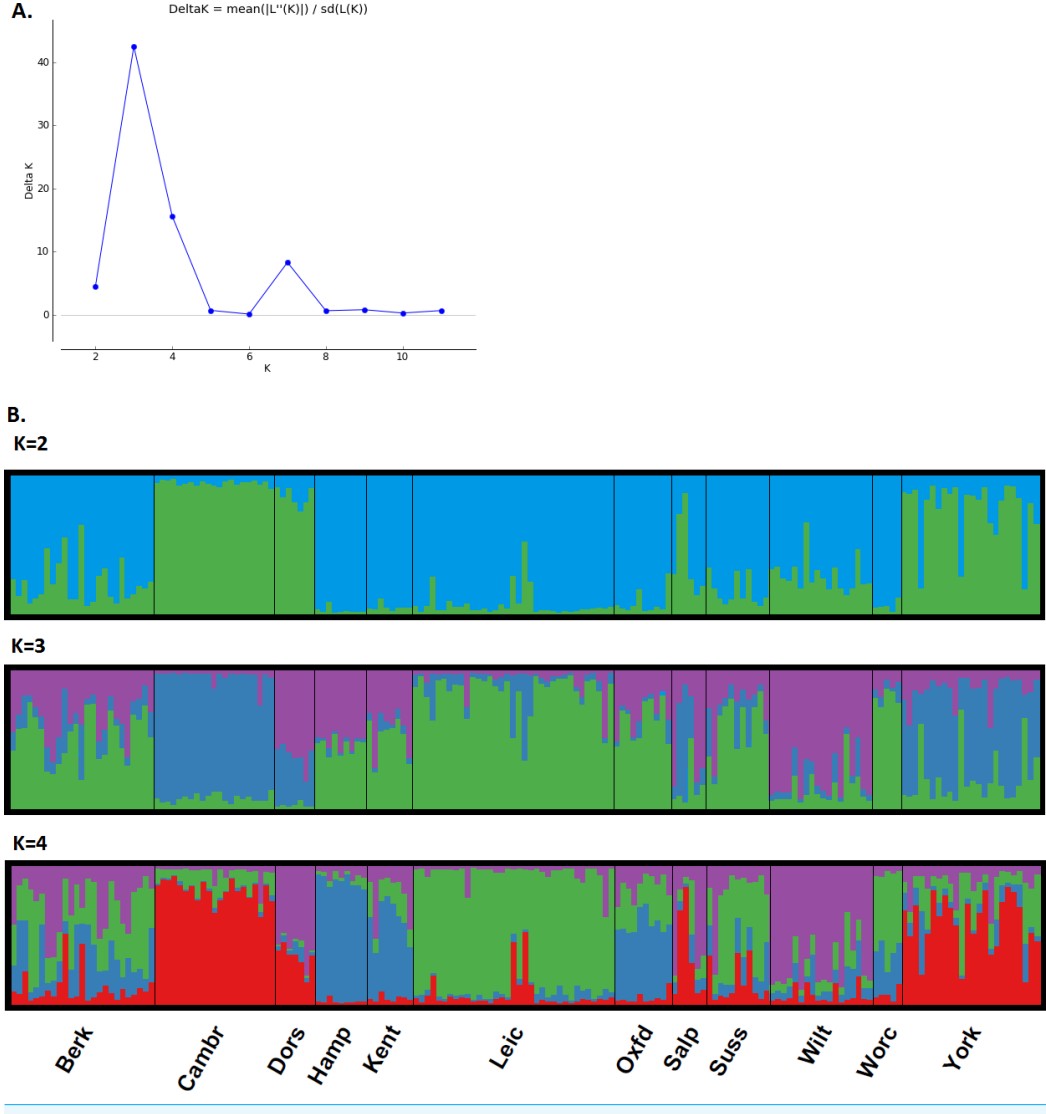

**Figure 4 Population structure of brown rats from microsatellite DNA data.** (A) shows the rate of change in the log-likelihood values between the number of genetic populations, $K$, for $K = 2$ to $K = 11$, showing that the value of $K$ with the greatest support is $K = 3$. (B) STRUCTURE plots of the average of ten runs of $K$ for $K = 2$, $K = 3$ and $K = 4$, showing the allocation of each individual (a single vertical bar) to each population, grouped by geographic sampling location along the horizontal axis. Each geographic location is divided by a thin black line, and the abbreviated geographic location is given at the bottom of $K = 4$.

## VKOR analysis

A result could not be obtained for two rats (Kent $n = 1$, Sussex $n = 1$), leaving a sample size of 184; VKORC1 variants were identified in 124 (67.4%) individuals (Table 5). The proportion of rats with VKORC1 variants varied between counties (Pearson's Chi-squared test: $X^2 = 29.59$, 13 $df$, $p = 0.005$) and the difference was apparently not due to sampling effort (Spearman's rank correlation: $r = -0.4$, $n = 14$, $p = 0.155$). The Leu128Gln variant was not found in any of our samples (the assay was checked using a positive control).

**Table 5  Distribution of VKORC1 variants by county.**

| County | N | No mutation (%) | Tyr139Cys | Leu120Gln | Tyr139Ser | Tyr139Cys, Leu120Gln | Tyr139Cys, Tyr139Ser | Tyr139Cys, Leu120Gln, Tyr139Ser |
|---|---|---|---|---|---|---|---|---|
| Berkshire | 25 | 3 (12.0) | 16 | 1 | 0 | 5 | 0 | 0 |
| Cambridgeshire | 21 | 8 (38.1) | 13 | 0 | 0 | 0 | 0 | 0 |
| Dorset | 7 | 3 (42.9) | 4 | 0 | 0 | 0 | 0 | 0 |
| Gloucestershire | 3 | 0 (0.0) | 2 | 0 | 0 | 1 | 0 | 0 |
| Greater London | 3 | 1 (33.3) | 2 | 0 | 0 | 0 | 0 | 0 |
| Hampshire | 9 | 2 (22.2) | 1 | 1 | 0 | 4 | 0 | 1 |
| Kent | 7 | 3 (42.9) | 4 | 0 | 0 | 0 | 0 | 0 |
| Leicestershire | 35 | 19 (54.3) | 16 | 0 | 0 | 0 | 0 | 0 |
| North Yorkshire | 24 | 13 (54.2) | 8 | 1 | 0 | 0 | 2 | 0 |
| Oxfordshire | 10 | 2 (20.0) | 7 | 0 | 0 | 1 | 0 | 0 |
| Powys | 2 | 1 (50.0) | 1 | 0 | 0 | 0 | 0 | 0 |
| Shropshire | 4 | 0 (0.0) | 2 | 0 | 0 | 0 | 2 | 0 |
| Sussex | 11 | 0 (0.0) | 8 | 1 | 0 | 2 | 0 | 0 |
| Wiltshire | 18 | 5 (27.8) | 11 | 2 | 0 | 0 | 0 | 0 |
| Worcestershire | 5 | 0 (0.0) | 5 | 0 | 0 | 0 | 0 | 0 |
| Total | 184 | 60 (32.6) | 100 | 6 | 0 | 13 | 4 | 1 |

Two VKORC1 variants were identified in seventeen individuals (Tyr139Cys + Leu120Gln, $n = 12$; Tyr139Cys + Tyr139Ser, $n = 4$; Leu120Gln + Tyr139Ser, $n = 1$); three variants were identified in a single individual (from Hampshire); Tyr139Cys + Leu120Gln + Tyr139Ser. There was an association between VKORC1 variant present and county ($G = 46.5$, $df = 26$, $p < 0.01$) but there was no association between variant present and the population structure suggested by the microsatellite data ($G = 14.9$, $df = 12$. N.S.).

A subsample of the rats (79) had been previously tested for susceptibility to warfarin by the blood clotting response (BCR) test (*Kerins et al., 2001*); four were found to be susceptible. However, in these four individuals the Tyr139Cys variant was identified; three of these individuals were from one farm in Berkshire, the fourth, in which the Leu120Gln variant was also identified, was from Gloucestershire. Furthermore, 10 of the 75 (BCR) resistant rats showed none of the four VKORC1 variants that we had initially screened for. We sequenced exon 3 of these 10 BCR positive rats; in six we found no VKORC1 variants, in two of the remaining four we found the Leu120Gln variant, and in the other two we found a Tyr139Phe amino acid substitution.

## DISCUSSION

We analysed genetic variation in rural brown rat populations in England using both mitochondrial and microsatellite DNA in order to investigate the population structure of this important pest species. In these populations, we also quantified the number and type of VKORC1 variants previously reported to be associated with resistance to warfarin in the UK.

For the mtDNA data, we found two haplogroups, RNH1 and RNH6, and RNH2, RNH4, RNH5 and RNH3. The divergence between these two haplogroups was quite high (six mutational positions) relative to the within group divergence (one-two mutational positions), suggesting that these two clusters represent distinct founding haplotypes rather than *in situ* divergence. There is no geographic structuring to the distribution of these haplogroups, suggesting either that they were introduced and spread concurrently, or that they were introduced in two stages but subsequently spread panmictically. This lack of mtDNA geographic structuring contrasts to that found in another rodent pest, the house mouse, which has geographic structuring in the UK (*Searle et al., 2009*), although this was detected at a wider scale than was found here.

In contrast, the STRUCTURE analysis of the microsatellite data suggested that there was some degree of geographic population structuring within the UK, with a best fit of three populations ($K = 3$). There is some geographic coherence to the distribution of these populations, indicating they are biologically relevant. A confounding variable is that there may also be a temporal element, as the Leicester, Cambridgeshire and Yorkshire samples were all collected in 2005 rather than 1993–1994; Cambridgeshire and Yorkshire form a distinct cluster. To compare the results again to house mice, studies in Ireland and France found the microsatellite data strongly matched the geographic origin of the samples (*Jones et al., 2011*).

Discrepancies of this kind between mitochondrial and microsatellite data have been found in numerous studies (*Waits et al., 2000*). These discrepancies can be attributed to differences in the levels of male and female gene flow (*Avise, 1994*), as the dispersal distance of male rats is greater than females (*Calhoun, 1962*). However, this would imply a greater degree of geographic structure shown by the (female dispersed) mtDNA data than that shown by the (male and female dispersed) microsatellite data. A more likely explanation for these results is that the relatively high mutation rate of microsatellites better reflects population structuring than the slower mutating mtDNA sequence, which is informative about earlier events. This is particularly likely to be the case as the fragment of mtDNA extracted here is relatively short.

The haplotype and nucleotide diversity of mtDNA in our samples is relatively low, although it is similar to the values found for the black rat on Madagascar (*Hingston et al., 2005*). We found only six haplotypes differing at 10 positions. Low nucleotide diversity found in a widespread species is often attributed to a slow range expansion following a small population size (founder/ bottleneck effects) (*Joseph, Wilke & Alpers, 2002*). Further support for this is given by the distribution of pairwise haplotype differences (Fig. 3), which matches that expected under a spatial expansion model (*Rogers & Harpending, 1992*).

If, as seems likely from our data, the English brown rat population has undergone a recent (on an evolutionary scale) expansion, it is unlikely to be yet at equilibrium. Regional equilibrium can be tested by comparing $F_{st}$ to geographical distance between regions (*Hutchison & Templeton, 1999*). If the population has reached equilibrium, there will be a linear relationship between $F_{st}$ and geographical distance. We found no significant

relationship between $\Phi_{st}$ (analogous to $F_{st}$) and geographical distance. Our results most closely resemble Hutchison & Templeton's case III, where the population is fragmented into small, isolated populations and drift becomes more important than gene flow. This allows allele frequencies in each population to drift independently relative to geographical distance and random sampling of gametes creates a large degree of variance between the plotted points (*Hutchison & Templeton, 1999*). We found no significant correlation between the residual of $\Phi_{st}$ and geographical distance (a measure of the degree of variance) and geographical distance, indicating that our data do indeed fit the case III model. This model and our data suggest that the English rural rat population is not yet at equilibrium and that gene flow is less important than drift in explaining the genetic structure found.

Accordingly, we found a geographical trend (by county) in the distribution of VKORC1 variants. The Leu120Gln amino acid substitution was found in the central and southern counties. The Tyr139Cys substitution was the most common and found in the majority in almost all counties; in Hampshire a combination of Tyr139Cys and Leu120Gln substitutions was found more frequently than Tyr139Cys alone, although the sample size for the county (nine) was relatively small. The Tyr139Cys substitution is reportedly better at ameliorating the effects of (and therefore more likely to confer a selective advantage against) warfarin use than Leu120Gln (*Pelz et al., 2005*), which potentially provides an explanation for the more widespread distribution of Tyr139Cys. The Tyr139Cys amino acid substitution is almost ubiquitous in Germany and Denmark (*Pelz et al., 2005*). We found only five instances of the Tyr139Ser variant; all of these were in combination with other VKORC1 amino acid substitutions. The Tyr139Ser variant has previously been reported from rats near the Anglo-Welsh borders around the town of Welshpool, and not from elsewhere in the UK (*Buckle, 2013*). Our two records of this VKORC1 variant from Shropshire were from rats collected near the town of Shrewsbury, 30 km from Welshpool. Our two records of this variant from Yorkshire, and the single record from Hampshire are noteworthy given their considerable distance from previous records, although it is possible that these represent false positives; *Mln* I digestion of the Tyr139Ser variant may produce fragments close in size (110 bp plus < 50 bp) to wild type alleles (160 bp plus < 50 bp) (*Pelz et al., 2005*).

We found a mis-match between BCR results and VKOR polymorphisms for some rats; Tyr139Cys amino acid substitution was identified in four apparently susceptible rats. *Pelz et al. (2005)* suggested that such cases were false negatives due to inaccuracies of the BCR test, although an alternative explanation is that the Tyr139Cys amino acid substitution alone is not sufficient to confer resistance, and an additional (undetected) substitution (or other physiological mechanism) is required which these four rats did not possess. We also found that six BCR positive rats had no VKORC1 amino acid substitutions, that is, they were BCR false positives, unless there is an as yet undiscovered, alternative resistance mechanism (*Pelz et al., 2005*), and a similar result has been reported elsewhere (*Heiberg, 2009*). The Leu120Gln variant was identified in two BCR positive rats that were initially thought to be false positives, whilst in a further two apparent BCR false positives, a Tyr139Phe amino acid substitution not known from the UK at the time of our study was

found, although this VKORC1 variant has since been reported in rats from Kent (*Prescott et al., 2010*). Several other newly-discovered VKORC1 variants have recently been reported from the UK (*Buckle, 2013*). We initially screened samples for the four VKORC1 variants reported from the UK at the time of our study, and did not look for other variants, except where BCR false positives were suspected. It is possible therefore that we underestimated the prevalence of VKOR polymorphisms in our samples. However, the more recently reported VKORC1 variants also appear to have limited (regional) distribution, which is consistent with our suggestion that limited gene flow between regions limits geographical distribution of VKOR polymorphisms.

It has been reported that because of pleiotropic costs (on vitamin K requirements, reduced fecundity and growth rate) that anticoagulant resistant rats (from some regions) are at a selective disadvantage in the absence of anticoagulant use compared to susceptible individuals (*Jacob et al., 2012*; *Smith, Townsend & Smith, 1991*). Intuitively therefore, these susceptible rats should out-compete their resistant counterparts in the absence of exposure to anticoagulant rodenticides, and it follows that resistance-management strategies should, where possible, include the use of non-anticoagulant rodenticides and non-rodenticide approaches in order to remove selection pressure, and to remove resistant individuals (*Buckle, 2013*; *Greaves, 1995*; *Lambert et al., 2008*; *Quy et al., 1995*; *Smith & Greaves, 1987*). However, in a captive, insular rat population, anticoagulant tolerance was not significantly influenced in the absence of bromadiolone selection (*Heiberg, Leirs & Siegismund, 2006*), and not all warfarin resistant strains appear to be at a selective disadvantage in the absence of poison use (*Smith et al., 1993*). In a field trial where removal of anticoagulant selection pressure from a population of rats highly resistant to bromadiolone did not result in a greatly increased proportion of susceptible individuals within the population, it was found that a high proportion of rats on surrounding farmsteads were also resistant to bromadiolone, and there was therefore little opportunity for susceptible rats to dilute the resistant population through immigration (*CSL, 2002*). The lack of mixing between populations at larger geographical scales, revealed by this study, is likely to intensify the effect of local selection pressure imposed by sustained anticoagulant use, and reversing these processes is therefore likely to be slow and difficult to achieve. However, the present study also suggests that restricted gene flow between rat populations should limit the rate of spread of resistant populations to some degree, and should therefore make their targeted management a realistic possibility. *Greaves (1995)* suggested that prompt and sustained control programmes using a suitable range of non-selective techniques within a 20 km radius of anticoagulant foci would be very likely to extinguish the majority of resistant populations. However, *Buckle (2013)* noted that such large-scale coordinated resistance management efforts have previously proven prohibitively expensive or impractical in the UK. Some of the most toxic second-generation anticoagulant rodenticides (SGARs) including brodifacoum and flocoumafen are still effective against rats resistant to warfarin, bromadiolone and difenacoum, and their use in areas where resistance to the less-toxic anticoagulants is encountered has been recommended where alternative (non-anticoagulant) methods cannot be used (*Buckle, 2013*). However, it is unknown

what the impacts of this approach will be on resistant rat populations in the longer-term, and it remains important that alternative rodent control approaches and rodenticides are developed, so that a range of viable options is available to manage populations of anticoagulant resistant rats.

## ACKNOWLEDGEMENTS

We thank Carl Smith for his involvement with the work, and two anonymous referees who provided useful feedback on an earlier draft. We are grateful to Geoff Butcher from the Babraham Institute for providing the additional samples from Cambridge.

### Funding

MZHH was funded by a Malaysian government scholarship. The funders had no role in study design, data collection and analysis, decision to publish, or preparation of the manuscript.

### Grant Disclosures

The following grant information was disclosed by the authors:
Malaysian government scholarship.

### Competing Interests

The authors declare there are no competing interests.

### Author Contributions

- Mohd Z.H. Haniza conceived and designed the experiments, performed the experiments, wrote the paper, prepared figures and/or tables, reviewed drafts of the paper.
- Sally Adams performed the experiments, reviewed drafts of the paper.
- Eleanor P. Jones and Mark S. Lambert analyzed the data, wrote the paper, prepared figures and/or tables, reviewed drafts of the paper.
- Alan MacNicoll contributed reagents/materials/analysis tools, reviewed drafts of the paper.
- Eamonn B. Mallon conceived and designed the experiments, performed the experiments, analyzed the data, wrote the paper, reviewed drafts of the paper.
- Robert H. Smith conceived and designed the experiments, reviewed drafts of the paper.

### Animal Ethics

The following information was supplied relating to ethical approvals (i.e., approving body and any reference numbers):

All research that involved the use of live animals was approved in advance by written confirmation from Animal and Plant Health Agency, York, UK Animal Welfare and Ethical Review Body (AWERB) or equivalent at the time of the study. Our AWERB membership has both internal and external members including lay representatives and experts in a variety of apposite areas such as veterinary surgery, statistics and animal welfare.

## DNA Deposition

The following information was supplied regarding the deposition of DNA sequences:
Genbank: DQ897633–DQ897638.

## Supplemental Information

Supplemental information for this article can be found online at http://dx.doi.org/10.7717/peerj.1458#supplemental-information.

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
