# Peer review of "Large-scale structure of brown rat (*Rattus norvegicus*) populations in England: effects on rodenticide resistance"

_PeerJ, doi:10.7717/peerj.1458_

## Round 0.1 · original submission · Major Revisions

You use county as a way of defining rat populations (eg l 196). While I can understand that using county might be a reasonable way to define geographical clusters, I guess administrative limits do not have any biological meaning for rat populations. It would be much more powerful to define clusters on the basis of the physical environment with direct impact on rat populations - e.g. dispersal.
The way one should use STRUCTURE to determine the number of groups has been discussed in Evanno et al. (2005). Justify the way you have estimated K with reference to the recent literature on this topic.

Table 2: Rattus norvegicus.

Ref: Evanno, G., S. Regnaut, and J. Goudet. 2005. Detecting the number of clusters of individuals using the software STRUCTURE: a simulation study. Molecular Ecology 14:2611-2620.

Reviewer 1 ·

Basic reporting

The articel is sufficient.

Experimental design

Some comments in terms of description and analysis are following in general cmments.

Validity of the findings

This paper covers the important topic of spread of rodenticide resistance in brown rats. I have some concerns (see general comments)

Additional comments

Dear Authors,

your paper covers the important topic of spread of rodenticide resistance in brown rats. My main concerns about this paper however are:

1) Why did you analyse only a 425 base pair region of mtDNA although other papers demonstrate, that an analysis of longer segments could be necessary (e.g., Robins et al. 2007)
2) Could results be improved by consideration of additional mtDNA segments and microsatellites?
3) Which role plays the different years of sampling?

Details:
Introduction: It should be presented more basic information about why you used both mtDNA and microsatellites.
p. 3 line 45: co-dominant not dominant
p. 4 line 75: year for Cambridge samples should be included
p. 5 line 112 (and elsewhere where numbers <10): four instead 4
p. 5 line 112: mutations should be named
p.7 Line 143: delete the point after ‘change’
figure 1: RNH2 and RNH6 have the same colour
p. 8 line 168: An example for what table 3 shows would be great. The same should be done for other figures and tables.
p. 8 line 182-183: The terms ‘observed heterozygosities’ (Ho) and ‘expected heterozygosities’ (He) should be explained in method part.
p. 12 line 250-259: Please explaine more in detail how differences between female and male in dispersal lead to the discrepancies between results from mitochondrial and microsatellite analysis.
p. 14 line 316: I cannot follow the assumption of out-competing!
p. 15 line 319: Why is Tyr139Ser a secondary mutation?
p. 15 line 324-335: What is the paragraph saying? Mutations can be find but no rodenticide resistance? I cannot see a connection with your analysis.
p. 15 line 342: It’s not really clear how the usage of warfarin in 1989 supported rodenticide resistance in the following years. It seems very vague for me, because of the spatial scale and because time scale effecta are not considered in the results.

Best regards.

Reviewer 2 ·

Basic reporting

The authors evaluated the interplay of population structure and anticoagulant rodenticide resistance in Norway rats in England. This is a topic of relevance for basic science and application.
The manuscript is mostly written and logical. However, there are 2 major issues with the experimental design (see below).

Experimental design

The study is based on samples taken from rats collected 1990-2005. This is a long period. A “snap shot” spanning 25 years with sampling finished 8 years ago may not reflect the current situation. This should be discussed and use of data justified.

The authors aim to correlate Warfarin use with the occurrence of “resistance mutations”. There is 1 data point for spatial Warfarin use in 1989. I don’t think these data are worth considering as the mutation data are spread across the following 25 years and suggest deleting related material in L 227-231, 337-355 and abstract.

Validity of the findings

I am a bit confused about the statement L 360 that susceptible rats are needed to dilute the presence of resistance mutations. In a management context – which is anticoagulant rodenticide use about – doesn’t one want to get rid of Norway rats no matter whether resistant or not?
There is no evidence for that in the manuscript anyway and findings of other studies are equivocal.

Additional comments

L 102 and refs is A.-C. HeibErg
L 132 provide ref
L 133 should read “1,000”
L 166-172 state main results instead of simply referring to Tabs and Figs
L 174 should read “R. rattus”
L 178 should read “1,000”
L 193-204 combine into 1 para
Abstract line 8 should read “populations.”

---

## Round 0.2 · Minor Revisions

The revision has addressed most issues raised previously. Only minor issues are left.

Reviewer 1 ·

Basic reporting

The articel is sufficient and meets the standard.

Experimental design

The experimental design is sufficient and meets the standard.

Validity of the findings

see general comments.

Additional comments

Dear Author,

The paper focuses on the mostly ignored topic of population structure in space of brown rats which is important for effective management.

I recommend this well written manuscript for publication with minor revisions.

My comments are:

If you speak from Tyr139Cys etc. then you speak from polymorphisms of the VKOR and you don’t speak from mutations of the VKORC1. This should be considered in the paper.

Line 67: The most important paper is Rost et al. 2004 (Mutations in VKORC1 cause warfarin resistance and multiple coagulation factor deficiency type 2. Nature 427: 537-541) and resistance effects are not only hypothesized.

Line 70: The mutation is co-dominant. Homozygosity leads to a stronger resistance effect than heterozygosity.

Line 83: ‘may be diluted’ instead of ‘will be diluted’

Line 146: ‘VKOR analysis’ instead of ‘mutation analysis’

Line 150-152: Due to you mention these polymorphisms, why you did not look for these mutations too? Add the reason.

Line 216-220: Should be shifted to methods/ Data analysis.

Line 292-305: Should be shifted to the discussion because comparison with results of another study.

Line 347-354: It is difficult to follow the intention. Section should be shorter focusing on the main statements and inclusive the section (line 292-305) from result part.

Line 401: I don’t understand ‘This represents a small error rate in our mutation detection....’. In which study it was false positive...? Please clarify this.

.
Regards

---

## Round 0.3 · accepted · Accept

The last minor issues were dealt with.